

# Plurality in multi-disciplinary research: multiple institutional affiliations are associated with increased citations

Paul Sanfilippo[1,2], Alex W. Hewitt[1,2,3] and David A. Mackey[1,2,3]

[1] University of Melbourne, Royal Victorian Eye and Ear Hospital, Centre for Eye Research Australia, Melbourne, Australia
[2] University of Western Australia, Lions Eye Institute, Centre for Ophthalmology and Visual Science, Perth, Australia
[3] School of Medicine, University of Tasmania, Menzies Institute for Medical Research, Hobart, Tasmania, Australia

## ABSTRACT

**Background**. The institutional affiliations and associated collaborative networks that scientists foster during their research careers are salient in the production of high-quality science. The phenomenon of multiple institutional affiliations and its relationship to research output remains relatively unexplored in the literature.
**Methods**. We examined 27,612 scientific articles, modelling the normalized citation counts received against the number of authors and affiliations held.
**Results**. In agreement with previous research, we found that teamwork is an important factor in high impact papers, with average citations received increasing concordant with the number of co-authors listed. For articles with more than five co-authors, we noted an increase in average citations received when authors with more than one institutional affiliation contributed to the research.
**Discussion**. Multiple author affiliations may play a positive role in the production of high-impact science. This increased researcher mobility should be viewed by institutional boards as meritorious in the pursuit of scientific discovery.

## INTRODUCTION

With the Digital Revolution, the time-honoured model of scientific discovery being contingent on a singular intellect working independently of others has expired. In the modern age of global travel and the interactive capabilities afforded by the internet, there is an expectation that good researchers are internationally mobile, both physically and virtually (*Schiermeier, 2011*). Researcher mobility is not a goal in itself, but rather a means of fostering collaborative networks at the many levels (e.g., institutional, interdisciplinary, international, etc.) that may drive successful scientific discovery. The increasing dominance of collaborative teams both within and between institutions has been documented to enhance efficiency and productivity as well as produce better science (*Wuchty, Jones & Uzzi, 2007*). This is also reflected in the growth of international teams and their association

Corresponding author
Paul Sanfilippo, prseye@gmail.com

with increased citation counts, a marker of research impact (*Wuchty, Jones & Uzzi, 2007*; *Jones, Wuchty & Uzzi, 2008*) Entangled within this collaborative research milieu, the institutional affiliations held by a researcher may also be viewed as a marker of capacity to facilitate knowledge exchange (*ESF, 2013*). However, to date there has been little research from the burgeoning scientometric and bibliometric fields exploring the role of multiple institutional affiliations on scientific output (*Hottenrott & Lawson, 2017*). To improve our understanding of this phenomenon, we conducted a large-scale analysis of scientific publications from four multi-disciplinary science journals (Science, Nature, Proceedings of the National Academy of Sciences (PNAS), PLOS Biology (PLOS)).

## MATERIALS & METHODS

We retrieved all 'articles' listed for the above journals from Web of Science (WoS) for the years 2010–2014, inclusive (search performed on 14/06/17). Articles were exported from WoS as BibTeX files, with complete metadata, then imported into the R statistical environment (*R Core Team, 2017*) for further processing. The bibliometrix package (*Aria & Cuccurullo, 2016*) was used to create a bibliographic data frame with cases (rows) corresponding to manuscripts and variables (columns) to Field Tags (metadata) in the original BibTex file. In this way the bibliographic attributes for each article (i.e., title, author's names, author's affiliations, citation count, document type, keywords, etc.) are formatted appropriately for subsequent analysis. The most important Field Tag for the purposes of this study is the Author Address (C1) tag which provides institutional address information for each author and where an author has multiple affiliations, lists these addresses separately. We split each manuscript record by author name and affiliation address, with the sum of author name occurrences indicating the number of distinct affiliations for that author. As comparisons of raw citation counts are biased by virtue of time since publication (i.e., earlier publications have had longer to accumulate citations), normalized citation counts were computed by dividing the raw value by the number of days since June 30th of the year of publication through to the search date (14/06/17), and then multiplying by 365 (*Uddin & Khan, 2016*). This enables unbiased comparisons of citation counts irrespective of the year of publication.

## RESULTS AND DISCUSSION

Of the 27,651 articles retrieved, 39 did not have affiliation data recorded and were excluded. The total number of articles available for analysis was 27,612, with Science ($n = 3,910$), Nature ($n = 4,120$), PNAS ($n = 18,651$), and PLOS ($n = 931$). The maximum number of citations for a single paper (published in 2012) was 4,143 (mean and median: 79.6 and 43.0, respectively). The maximum number of normalized citations was 828, for the same paper (mean and median: 15.7 and 8.8, respectively). The maximum number of authors for a single paper was 2,908 (mean and median: 9.0 and 6.0, respectively), and the maximum number of author affiliations was 271 (mean and median: 4.7 and 4.0, respectively). Author affiliations were recorded as presented by WoS.

**Table 1 Frequency distribution of articles and author appearances in most- and least-cited articles, stratified by the number of author affiliations attached to each article.** As individual articles may have contained multiple authors with different numbers of affiliations, they may appear more than once in the summary (i.e., an author may appear on multiple papers). Consequently, the values do not represent *unique* numbers of articles or authors. Highest citations = normalized citations > 8.8 (unique articles = 13,795), Lowest citations = normalized citations ≤ 8.8 (unique articles = 13,817]).

| Number of affiliations | Number of article appearances | | Number of author appearances | | |
|---|---|---|---|---|---|
| | Lowest citations | Highest citations | Lowest citations (%) | Highest citations (%) | Total (%) |
| 1 | 13,102 | 13,118 | 73,430 (29.4) | 111,750 (44.7) | 185,180 (74.1) |
| 2 | 7,327 | 8,803 | 19,174 (7.7) | 30,775 (12.3) | 49,949 (20.0) |
| 3 | 2,451 | 3,283 | 4,381 (1.7) | 6,718 (2.7) | 11,099 (4.4) |
| 4 | 640 | 1,027 | 1,012 (0.4) | 1,622 (0.7) | 2,634 (1.1) |
| 5 | 185 | 319 | 304 (0.1) | 457 (0.2) | 761 (0.3) |
| 6 | 46 | 72 | 51 (<0.1) | 109 (<0.1) | 160 (<0.1) |
| 7 | 8 | 25 | 8 (<0.1) | 29 (<0.1) | 37 (<0.1) |
| 8 | 7 | 6 | 7 (<0.1) | 7 (<0.1) | 14 (<0.1) |
| 9 | 0 | 2 | 0 | 8 (<0.1) | 8 (<0.1) |
| 10 | 0 | 1 | 0 | 1 (<0.1) | 1 (<0.1) |
| 11 | 0 | 0 | 0 | 0 | 0 |
| 12 | 0 | 1 | 0 | 2 (<0.1) | 2 (<0.1) |
| Total | | | 98,367 (39.4) | 151,478 (60.6) | 249,845 (100) |

Table 1 shows the distribution of article and author appearances stratified by the number of author affiliations for the most- and least-cited articles split at the median normalized citation value (Highest citations = citations > 8.8 [$n = 13,795$], Lowest citations = citations ≤ 8.8 [$n = 13,817$]). While the vast majority of author appearances were associated with only one institutional affiliation (74.1%), 25.9% of author appearances were linked with two (20.0%) or more affiliation addresses. The maximum number of institutional affiliations held by an author was 12. As these are non-independent observations, classical tests of contingency tables are not appropriate; however, one can easily appreciate the increased frequency of author appearances in the more-cited publications. Indeed, the correlation between the normalized number of citations a paper received and the number of authors on that paper was statistically significant ($\rho = 0.17$, $p \leq 0.001$). Similarly, the correlation coefficient for the normalized citations a paper received and the number of institutional affiliations on that paper was 0.25, $p \leq 0.001$. The correlation between the number of authors and number of affiliations listed for each paper was greater, indicating closer correspondence between the variables (0.67, $p \leq 0.001$).

To facilitate a simple yet fruitful investigation of the relationship between the number of normalized citations a paper received and its association with authorship and affiliation frequency, we categorised the latter two variables. The number of authors attached to each paper was split into quartiles to create an 'Author Number' variable, with the following categories: 1 = 1–3 authors/article, 2 = 4–5 authors/article, 3 = 6–9 authors/article, and 4 = 10–2,908 authors/article. Due to the low cell counts (Table 1) and to improve estimation

**Table 2 Frequency distribution (%) of unique articles in each category of Author Number and Maximum Affiliation.** Maximum Affiliation is the maximum number of affiliations held by a single author for each article, whilst the Author number is the number of authors per article.

| Author Number | Maximum Affiliation | | | | | | Total (%) |
|---|---|---|---|---|---|---|---|
| | **1** | **2** | **3** | **4** | **5** | **6** | |
| 1–3 | 3,142 (11.40) | 1,371 (4.97) | 454 (1.65) | 103 (0.37) | 24 (0.09) | 4 (0.01) | 5,098 (18.49) |
| 4–5 | 2,715 (9.85) | 2,207 (8.01) | 811 (2.94) | 210 (0.76) | 61 (0.22) | 9 (0.03) | 6,013 (21.81) |
| 6–9 | 2,898 (10.51) | 3,845 (13.95) | 1,509 (5.47) | 419 (1.52) | 119 (0.43) | 35 (0.13) | 8,825 (32.02) |
| **>9** | 1,387 (5.03) | 3,374 (12.24) | 1,859 (6.74) | 695 (2.52) | 250 (0.91) | 64 (0.23) | 7,629 (27.68) |
| Total (%) | 10,142 (36.79) | 10,797 (39.17) | 4,633 (16.81) | 1,427 (5.18) | 454 (1.65) | 112 (0.41) | 27,565 (100.00) |

in subsequent modelling, the maximum number of author affiliations held on a single paper was limited to six. This resulted in the exclusion of a further 47 papers, with 27,565 articles available for analysis. 'Maximum Affiliation' represents the maximum number of institutional affiliations held by a single author on an article. For example, if WoS listed an article with three authors each having two affiliations, and two authors each having three affiliations, in this case maximum affiliation would equal three. Table 2 shows the frequency distribution of articles by author number and maximum affiliation.

Figure 1 shows boxplots of citation counts for each category of author number and maximum affiliation. There is a general trend of normalized citation count increasing across both factors. We explored this relationship further in a linear regression model with normalized citation count as the outcome, and author number and maximum affiliation as predictor variables (Table S1). Although these are technically count data, the mean citation value is high and the distribution of the count model approximates the normal. Consequently, we have considered citations a continous variable and utilised a linear model. We initially fit a model with an interaction term (author number × maximum affiliation) and evaluated its signficance with a Wald test. The resulting $p$-value was highly significant (<0.001) suggesting the 15 coefficients for the interaction terms are not simultaneously equal to zero, and an interaction effect exists between the two variables (i.e., the relationship between maximum affiliation and citations received, varies depending on the value of author number). The model was checked for multicollinearity using the generalized variance inflation factor (GVIF). The raw output from the regression model is supplied in the Table S1. As interaction terms make coefficient interpretation difficult, results for the effect of each level of predictor are presented in a stratified manner, while holding the other predictor constant (Table 3). In addition, we adjusted for year of publication and journal in the analysis. It is of interest to note the effect of journal on normalized citation counts. Using PNAS as the reference category journal (chosen as the most populous), both Science and Nature receive on average higher normalized citation

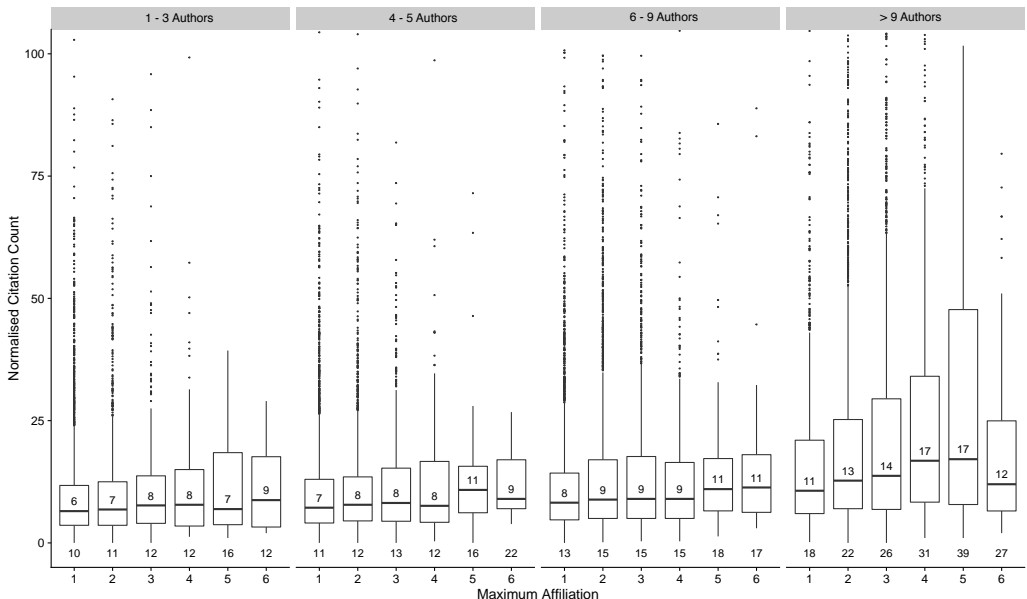

**Figure 1 Boxplots of citation counts stratified by author number and maximum affiliation** The horizontal line and adjacent number indicate the median, the top and bottom of the boxes the interquartile range, and the number below each plot, the mean citation count. Citations are truncated at 500.

counts per paper ($p < 0.001$) in comparison. Citations received were not significantly different between PNAS and PLOS.

Table 3 shows the effect for each combination of maximum affiliation and author number on normalized citation count. To further facilitate interpretation, we have limited maximum affiliation data to four addresses. The effect size (average change in normalised citation count) was computed using a series of linear contrasts that enables the comparison of differences among coefficients beyond the standard regression output. There are two main findings from these data: first, the effect on citation count of an author holding more institutional affiliations increases as the number of authors on a paper grows; and second, increasing the number of authors on a paper tends to result in more citations received irrespective of the number of affiliations held.

When there are between one to five authors/article, increasing the number of affiliations an author holds (relative to one) does not affect the average change in citation count. However, when there are between six to nine authors/article, authors with two institutional affiliations (relative to one) will, on average, increase the citations a paper receives by 1.6 ($p = 0.006$). This effect is even more pronounced when there are more than nine authors listed; here, citations increase on average by 2.3 ($p = 0.002$) for two affiliations, 5.8 ($p < 0.001$) for three affiliations and 9.4 ($p < 0.001$) for four affiliations, relative to the reference group.

If we now interpret these effects while holding the number of affiliations constant, for researchers with only one affiliation, increasing the number of authors on a paper results in a mean increase in the citations received across all levels of author number (e.g., 6.5 for
**Table 3  Summary of regression model output for the effect of Author Number and Maximum Affiliation on average citation counts.** Within each stratum, the average change in citation count is relative to the first (reference) level.

| Covariate | Effect | Average normalised citation count | Average change in normalised citation count | 95% CI for average change | *P* |
|---|---|---|---|---|---|
| Author Number = 1 | Max. Affiliation = 1 | 15.4 | 0 | | |
| (1–3 authors/article) | 2 | 15.8 | 0.4 | −1.1–1.9 | 0.60 |
| | 3 | 16.9 | 1.5 | −0.8–3.8 | 0.20 |
| | 4 | 18.9 | 3.5 | −1.1–8.0 | 0.14 |
| Author Number = 2 | Max. Affiliation = 1 | 16.7 | 0 | | |
| (4–5 authors/article) | 2 | 17.2 | 0.5 | −0.8–1.8 | 0.46 |
| | 3 | 18.1 | 1.4 | −0.4–3.2 | 0.13 |
| | 4 | 18.2 | 1.5 | −1.8–4.8 | 0.37 |
| Author Number = 3 | Max. Affiliation = 1 | 17.7 | 0 | | |
| (6–9 authors/article) | 2 | 19.3 | **1.6** | **0.5–2.7** | **0.006** |
| | 3 | 19.7 | **2.0** | **0.5–3.4** | **0.009** |
| | 4 | 19.6 | 1.9 | −0.5–4.3 | 0.11 |
| Author Number = 4 | Max. Affiliation = 1 | 21.9 | 0 | | |
| (>9 authors/article) | 2 | 24.2 | **2.3** | **0.8–3.7** | **0.002** |
| | 3 | 27.7 | **5.8** | **4.2–7.4** | **<0.001** |
| | 4 | 31.3 | **9.4** | **7.2–11.5** | **<0.001** |
| Max. Affiliation = 1 | Author Number = 1 | 15.4 | 0 | | |
| | 2 | 16.7 | **1.3** | **0.02–2.4** | **0.05** |
| | 3 | 17.7 | **2.3** | **1.1–3.5** | **<0.001** |
| | 4 | 21.9 | **6.5** | **5.0–7.9** | **<0.001** |
| Max. Affiliation = 2 | Author Number = 1 | 15.8 | 0 | | |
| | 2 | 17.2 | 1.4 | −0.3–2.9 | 0.10 |
| | 3 | 19.3 | **3.5** | **2.1–4.9** | **<0.001** |
| | 4 | 24.2 | **8.4** | **6.9–9.8** | **<0.001** |
| Max. Affiliation = 3 | Author Number = 1 | 17.0 | 0 | | |
| | 2 | 18.1 | 1.1 | −1.6–3.8 | 0.42 |
| | 3 | 19.7 | **2.7** | **0.3–5.2** | **0.03** |
| | 4 | 27.7 | **10.7** | **8.4–13.2** | **<0.001** |
| Max. Affiliation = 4 | Author Number = 1 | 18.9 | 0 | | |
| | 2 | 18.2 | −0.7 | −6.2–4.8 | 0.80 |
| | 3 | 19.6 | 0.7 | −4.2–5.8 | 0.76 |
| | 4 | 31.3 | **12.4** | **7.6–17.2** | **<0.001** |

author number = 4, relative to 1, $p < 0.001$). However, this effect remains significant for only greater author numbers (i.e., four vs. one) as the maximum number of affiliations held, increases. We would like to remind the reader that these data are cross-sectional in nature, and our discussion of 'effects' in the context of regression analysis does not imply causation in the relationships explored.

## CONCLUSIONS

These data align with previous observations in highlighting the increasing leverage of teamwork in scientific research (*Wuchty, Jones & Uzzi, 2007*; *Jones, Wuchty & Uzzi, 2008*). They also serve to provide some insight into the relatively novel notion that multiple author affiliations may play a positive role in the production of high-impact science (*Hottenrott & Lawson, 2017*). However, longitudinal analyses of citation count data would be necessary to explore the basis for a causal relationship. To that end, further research is needed to address some of the questions arising from the main finding of this study. What causes multi-institutional, larger authored papers to have greater citation impact? Is increased institutional representation seminal in the generation of high-quality science and therefore more highly cited works? Or are we observing an artefact of highly-funded and highly-competitive research that by its nature will generate more citations, irrespective of the number of authors or their affiliations. Clearly more data are needed to comprehensively address these points. Until then, the holding of multiple affiliations by authors should be viewed by institutional boards as a virtue and not a vice, as it appears that greater researcher mobility may be advantageous to all.

### Funding

The Centre for Eye Research Australia receives Operational Infrastructure Support from the Victorian Government. Paul Sanfilippo and Alex W. Hewitt are supported by NHMRC Fellowships. The funders had no role in study design, data collection and analysis, decision to publish, or preparation of the manuscript.

### Grant Disclosures

The following grant information was disclosed by the authors:
Victorian Government.
NHMRC Fellowships.

### Competing Interests

The authors declare there are no competing interests.

### Author Contributions

- Paul Sanfilippo conceived and designed the experiments, performed the experiments, analyzed the data, prepared figures and/or tables, authored or reviewed drafts of the paper, approved the final draft.
- Alex W. Hewitt and David A. Mackey conceived and designed the experiments, performed the experiments, authored or reviewed drafts of the paper, approved the final draft.

### Data Availability

Sanfilippo, Paul (2018): Plurality.RData. figshare. Dataset. Available at https://doi.org/10.6084/m9.figshare.7033631.v1

## Supplemental Information

Supplemental information for this article can be found online at http://dx.doi.org/10.7717/peerj.5664#supplemental-information.

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
