# Peer review of "Plurality in multi-disciplinary research: multiple institutional affiliations are associated with increased citations"

_PeerJ, doi:10.7717/peerj.5664_

## Round 0.1 · original submission · Major Revisions

The three reviewers raised multiple relevant points regarding the background and methodology of the work. Potential biases of citation (as raised by Reviewers 1 and 2 and alluded to by reviewer 3) for newer vs. older papers are particularly concerning. Reviewers 2 and 3 raise concerns about the motivation and the contextualization in prior work, and both also raise significant questions regarding the presentation and interpretation of the work. Please address these comments in any resubmission.

Reviewer 1 ·

Basic reporting

Okay, some refinement with table illustration (i.e., table 1) would make it even better

Experimental design

Research question is well defined and is a relevant one. Basic statistical methods were followed for research analysis.

Validity of the findings

The result is not robust. In fact, it is significantly biased. The authors need to use a normalised value for the citation count. An article published in 2005 (for example) will obviously get more time to be cited compared to another article published in 2015.

Additional comments

Without considering the normalised value of citation count the entire findings of this research will be completely misleading. I would recommend the authors to consider the following article regarding how to normalise a citation count value:

The impact of author-selected keywords on citation counts
By S Uddin, A Khan, Journal of Informetrics 10 (4), 1166-1177

·

Basic reporting

Background/context could be improved (see suggestions below).

Experimental design

This article looks at the correlation between author numbers, affiliation numbers and citations. In doing so it fits into the broader range of articles looking at the importance of network size on citation impact.

I have several comments:

1. The title is misleading and should be changed: a) The title refers to ‘biomedical research’ but the authors analyse multi-disciplinary journals which cover a wide range of different subject areas, not just biomedical. b) The notion of ‘improved research output’ is misleading. The article looks at citations, and this should be reflected in the title. c) Author numbers are more consistently associated with citations, while multiple affiliations only under limited circumstances show such association. Thus, the title gives a very wrong impression of the findings.

2. The paper is poorly motivated. I understand that it wants to contribute to the study of the importance of teams in knowledge production. It does this by adding the dimension of multiple affiliations to the analysis of co-author numbers. This is all fine, but the whole introduction about digital revolution and mobility is entirely misplaced. More effort should also be made to tell us what the papers that are cited found, such as the increase in team sizes [1] and increase in multiple affiliations [4], as well as the growth of international teams and their association to citation numbers [2]. At present the research question and knowledge gap are not clear.

3. The paper does not mention how citations are counted. Are they looking at all citations received by 14/06/2017? Having citations at one point in time means that 2010 publications had 7 years to accumulate citations, while 2014 publications had just 3 years. This truncation may influence the findings if the number of authors or affiliations has increased over the same period (something that the references [1] and [4] suggest). This truncation problem is normally addressed through the use of fixed citation windows. Moreover, the regressions analysis (supplementary table) should control for year and journal as both may affect citation counts.

4. I find the discussion of the results somewhat confusing, especially as no complete descriptive statistics and correlation tables are provided. Table 1 specifically should provide percentages in addition to paper numbers and the table caption should include a definition of ‘lowest’ and ‘highest’ citations. Table 3 in turn is difficult to read as the average number of citations is not known. This makes it difficult to understand what the citations difference actually means (i.e. whether it is small or large).

Validity of the findings

5. The article uses the language of causality, when all they provide is correlations. For instance on lines: 97, 119, 135 phrases such as ‘influence’, ‘effect’, ‘affect’ are used. This is plainly unacceptable as no conclusions regarding the direction of this correlation can be made. It is not clear whether better, more cited author attract more affiliations and co-authors or whether more affiliations or coauthors automatically lead to better, more cited research.

6. The conclusion that multiple affiliations ‘may also play a positive role in the production of high-impact science’ (152-153) is thus very tentative and can hardly said to be supported here. Longer time-series and an ex-ante/ex-post comparison of citations would be necessary to approach this topic. The one interesting finding one can read from your analysis is that team size and multiple affiliations seem to be complementary. This is what we can learn here and what could be stressed in the conclusions.

Reviewer 3 ·

Basic reporting

I have a few initial concerns upon reading the abstract and introduction:

1. First, the background suggests there is no prior research regarding multi-institutional research teams and output productivity. But yet the social sciences (e.g., team science), and particularly network sciences, have been publishing citation/bibliometric analysis for quite some time. I think your claim is that they haven’t analyzed multi-institutional teams as a factor, yet the Jones et al [2] and Hottenrott [4] papers have indeed. Specifically, there are over 80 citations available from a Science of Team Science research community reference list on this issue, which I have attached ( “Authorship, Contributorship, Publishing Issues” folder of the Science of Team Science (SciTS) group in Mendeley).
2. Second, the use of the term polygamous behavior seems inflammatory and suggestive of illicit research. Why this choice of term?
3. Third, the suggestion that multi-institutional research is shunned by IRBs is generally not true (though it may be true on some campuses). It complicates the review process, but I don’t think there is evidence that IRBs prevent this kind of research. What is the basis for your suggestion that IRBs shun multi-I research? This is not discussed or reviewed at all in the paper,

[2] Jones BF, Wuchty S, & Uzzi B (2008) Multi-university research teams: shifting impact, geography, and stratification in science. in Science (American Association for the Advancement of Science), pp 1259-1262.
[4] Hottenrott H & Lawson C (2017) A first look at multiple institutional affiliations: a study of authors in Germany, Japan and the UK. in Scientometrics (Springer Netherlands), pp 285-295.

Experimental design

1. Jones et al used at dataset that included “4.2 million papers published over three decades.” Why did you choose 27,000 papers in a 4-year period? What was the justification of this smaller dataset?
2. Given the data ends 3 years prior to analysis, is that sufficient time to realize citation impact differences?
3. Why the choice of the 4 journals chosen (e.g., these are the 4 highest impact journals in biomedical research)? Does that choice skew the results in any way?
4. Given only 6% of the articles in the data were multi-institutional, are you concerned about the possible insufficiency of data available to assess the contribution of multi-institutional affiliation? Why not create a larger data set and focus only on multi-institutional papers?

Validity of the findings

1. According to Table 1, 74% of articles in the dataset were authored by a single institution, and 20% by two institutions. Thus only 6% of the dataset was more than 2 institutions. Given your literature review of growing multi-institutional research, was this surprising?
2. Table 3 (and in text) – what are the units in the average change in citation count column? Assuming percent change, but it could be absolute value as well.
3. Table 3 – one of 2 main findings is that “increasing the number of authors on a paper tends to results in more citations irrespective of the number of affiliations held.” This is how I interpret this as well, which is not included in the Discussion.
4. Supplementary table – is the author number here the categorized sets (2, 3, and 4), or the raw number of authors? I think the former, but it’s not clear from the table presentation
5. Regression (and supplementary table) suggests that the significant predicting value is still the number of authors, not the breadth of institutional affiliation.
6. The overall finding is that when papers have fewer than 6 authors, the number of institutions doesn’t matter, but that for larger authored papers it does enhance the impact of the paper. What does this reflect? Is it the diversity and breadth introduced to large research teams, or that papers generated from much larger research teams tend to reflect quality and prestige in other ways?
7. The Conclusions section needs to include more substantial discussion of the results here. (see questions below)

Additional comments

1. Were you surprised that the majority of articles in that dataset were NOT multi-institutional? How does this compare to the suggestions from [2[ and [4] about a growth in multi- institutional research?
2. What is different in the findings of your study compared to what Jones et al and Hottenrott found?
3. What is the importance of 5+ authors being more likely to be multi-institutional and in turn have higher citation impact? Is this an artifact of highly-funded institutional research (e.g. NSF collaboration grants) that are highly competitive and therefore likely to generate high quality publications? Or is there something important about broadening the institutional representation to the quality of knowledge generated (which presumably leads to higher impact)? Neither of these possible explanations are discussed, and so we’re left with the question of what to do with the findings.

---

## Round 0.2 · Minor Revisions

Reviewer 1 raises the concern that you have not clearly specified the mechanisms of your normalization, and reviewer 2 requests that you add some references to accompany some currently uncited assertions. Please make these changes.

Reviewer 1 ·

Basic reporting

okay

Experimental design

In the rebuttal letter, the authors mentioned that they considered the normalised citation count values. They also mentioned that they followed the approach proposed by Uddin and Khan (2016). However, I did not find any details of that approach in the main manuscript file. On top of that, I did find any correct reference of this kind of normalisation. They just mentioned they used normalised citation count value but did neither explain nor provide reference regarding how did they do it.

Validity of the findings

Please see my comment in the previous section

Additional comments

In the rebuttal letter, the authors mentioned that they considered the normalised citation count values. They also mentioned that they followed the approach proposed by Uddin and Khan (2016). However, I did not find any details of that approach in the main manuscript file. On top of that, I did find any correct reference of this kind of normalisation. They just mentioned they used normalised citation count value but did neither explain nor provide reference regarding how did they do it.

·

Basic reporting

The background section still has too many unreferenced sentences. For example, the authors talk about 'expectations' but do not back this up.

It is not clear what 'roaming' is to mean in this context.

Experimental design

No comment

Validity of the findings

No comment

---

## Round 0.3 · accepted · Accept

Thank you for addressing the final reviewers' concerns.